# Isolation and characterization marine bacteria capable of degrading lignin-derived compounds

Peng Lu[1☯], Weinan Wang[2☯], Guangxi Zhang[2], Wen Li[2], Anjie Jiang[2], Mengjiao Cao[2], Xiaoyan Zhang[2], Ke Xing[2], Xue Peng[2], Bo Yuan[2‡*], Zhaozhong Feng[2‡*]

**1** College of Life Sciences, Anhui Normal University, Wuhu, China, **2** School of Life Sciences, Jiangsu Normal University, Xuzhou, China

☯ These authors contributed equally to this work.
‡ These authors also contributed equally to this work.
* boyuan@jsnu.edu.cn (BY); fzz2012@jsnu.edu.cn (ZF)

**Data Availability Statement:** All relevant data are within the paper and its Supporting Information files.

**Funding:** This study was supported by National Natural Science Foundation of China (31800327,

## Abstract

Lignin, a characteristic component of terrestrial plants. Rivers transport large amounts of vascular plant organic matter into the oceans where lignin can degrade over time; however, microorganisms involved in this degradation have not been identified. In this study, several bacterial strains were isolated from marine samples using the lignin-derived compound vanillic acid (4-hydroxy-3-methoxybenzoic acid) as the sole carbon and energy source. The optimum growth temperature for all isolates ranged from 30 to 35°C. All isolates grew well in a wide NaCl concentration range of 0 to over 50 g/L, with an optimum concentration of 22.8 g/L, which is the same as natural seawater. Phylogenetic analysis indicates that these strains are the members of *Halomonas*, *Arthrobacter*, *Pseudoalteromonas*, *Marinomonas*, *and Thalassospira*. These isolates are also able to use other lignin-derived compounds, such as 4-hydroxybenzoic acid, ferulic acid, syringic acid, and benzoic acid. Vanillic acid was detected in all culture media when isolates were grown on ferulic acid as the sole carbon source; however, no 4-hydroxy-3-methoxystyrene was detected, indicating that ferulic acid metabolism by these strains occurs via the elimination of two side chain carbons. Furthermore, the isolates exhibit 3,4-dioxygenase or 4,5-dioxygenase activity for protocatechuic acid ring-cleavage, which is consistent with the genetic sequences of related genera. This study was conducted to isolate and characterize marine bacteria of degrading lignin-derived compounds, thereby revealing the degradation of aromatic compounds in the marine environment and opening up new avenues for the development and utilization of marine biological resources.

## Introduction

Lignin, a major component of vascular plant tissue, is the second most abundant natural substance in nature after cellulose [1]. It plays a protective role against pathogenic and saprophytic organisms by guarding cellulose polymers from hydrolytic attack. Furthermore, it also allows

31570028), the project funded by Grant from Anhui National Natural Science Foundation (1708085QC67), the Priority Academic Program Development of Jiangsu Higher Education Institutions (PAPD), Postgraduate Research and Practice Innovation Program of Jiangsu Province, China (KYCX18_2138, KYCX17_1618). The authors declare that they have no conflicts of interest with the contents of this article.

**Competing interests:** The authors have declared that no competing interests exist.

for terrestrial plant growth by providing rigidity in the stems and supports the circulation of sap by waterproofing the vascular tissue [2].

The complex aromatic lignin heteropolymer is formed when crosslinked lignin monomers, such as guaiacyl, syringyl, and *p*-hydroxyphenyl, are further polymerized via β-aryl ether linkages, biphenyl bonds and heterocyclic linkages, by oxidase radicals. Although this molecule is highly recalcitrant, several enzymes with the ability to depolymerize lignin have been identified, such as lignin peroxidases (LiPs), manganese peroxidases (MnPs) and laccases [1]. These extracellular fungal enzymes attack lignin substructures in decaying plant cell walls, thereby releasing different lignin-derived breakdown products [2].

Vanillic acid, ferulic acid, *p*-hydroxybenzoic acid, syringic acid, benzoic acid and biphenyl are known lignin-derived compounds [3]. Detailed studies have been performed on the degradation of these low molecular compounds by soil bacteria [4, 5]. Research suggests that protocatechuic acid (PCA) is a key intermediate in the metabolism of lignin-derived compounds, resulting in aromatic ring cleavage. Three PCA metabolic pathways have been reported: 2,3-cleavage [6], 3,4-cleavage and 4,5-cleavage [7, 8]. The ring cleavage products from each pathway eventually enter the tricarboxylic acid cycle.

Rivers transport large amounts of vascular plant organic matter into the oceans, resulting in a carbon flux that is estimated to make up approximately 0.7% of terrestrial primary production [9]. The fact that lignin is only produced by terrestrial plants, and because it is a highly stable compound, makes it an excellent biomarker for the presence of terrestrial organic matter in lacustrine and marine environments [10]. Furthermore, when buried in sediments, this biomarker can aid in assessing the impact of land composition on marine environments [11, 12]. Marine microbes from the sea that are capable of degrading lignin potentially contribute to organic matter cycling [13]. Additional information can be inferred by determining the ratios of lignin-derived compounds. For instance, the synringyl/ vanillyl ratio, and the cinnamyl/ vanillyl phenols ratio can indicate the vegetative source; whereas the vanillic acid/ vanillin ratio and the syringic acid/ syringaldehyde ratio indicates the degradation state of lignin materials [14]. Research suggests that the average cycling time of dissolved lignin in marine environments is approximately 90 years [15], thereby highlighting the importance of understanding the degree and mechanisms of lignin degradation in the oceans, which, in turn allows, for a better understanding of carbon cycling within the marine environment.Some studies have examined lignin degradation by mixed microorganisms in marine environments [16, 17]; however, the microorganisms responsible for the degradation were not identified.

In this study, we isolated several bacterial strains from marine samples using the lignin-derived compound vanillic acid as the sole carbon and energy source. We then investigated the physiological characteristics of the isolates to determine their role in lignin mineralization in the marine environment.

## Materials and methods

All experiments in the materials and methods were conducted according to protocols approved by Jiangsu Normal University and Anhui Normal University and completed by researchers of the manuscript. Field sampling is funded by the National Natural Science Foundation of China, and belongs to the research content of the project. The study area is not a special marine protected area and does not involve endangered or protected species.

### Study site and sample collection

Seawater samples were collected from Lianyungang and sediments samples were collected from the East China Sea in Zhoushan, Zhejiang (30°58′20.71″N, 122° 95 41′22.10″E), People's

Republic of China. The depth of the sediment sample was 10 m below sea level, the water temperature was 20°C. Further, the sediments samples were placed in sealed plastic boxes surrounded with ice and brought to the laboratory within 24h.

## Chemicals and material

4-hydroxybenzoic acid (99.9%purity), benzoic acid (99.9% purity) and syringic acid (99.8% purity) were purchased from Sangon Biotech (Shanghai, China) at the highest purity available. Methanol (99.9%purity), trifluoroacetic acid (99.5%purity), and acetonitrile (99.9%purity) were purchased from Sigma-Aldrich (St. Louis, MO). Vanillic acid (99.9%purity), ferulic acid (99.9%purity) and protocatechuic acid (99.9%purity) were purchased from TCI Fine Chemicals (Tokyo, Japan). Marine Broth 2216 (DSMZ medium 604) was used as the nutrient medium. ONR7a medium (DSMZ medium #950, DSMZ, Germany), which contains only inorganic ions, was used as the defined seawater medium [18].

## Isolation and characterization of vanillic acid degraders

Both the direct plating method and the enrichment technique were used to isolate vanillic acid-utilizing bacteria. The autoclaved medium was allowed to cool at room temperature and the filter sterilized vanillic acid was added into the medium. Briefly, direct plating was carried out on ONR7a agar plates amended with 0.2% vanillic acid. Cultures were incubated at 25°C until colonies appeared. The individual colonies that utilized vanillic acid as the sole carbon source for growth were selected and purified by re-streaking 3 times. The enrichment technique was performed in liquid ONR7a media amended with 0.2% vanillic acid. Cultures were incubated with shaking at 25°C until turbid. They were then plated on ONR7a agar after repeating this enrichment technique 5 times to obtain pure colonies [19].

To isolate the total DNA, each isolate was grown to exponential phase in 10 ml Marine Broth at 25°C. The resulting cells were washed twice with TES buffer (pH 8.0) containing 0.1 M NaCl, 10 mM Tris-HCl, and 1 mM EDTA and then suspended in 2.5 ml of the same buffer amended with 5 mg of lysozyme. After being kept at 37°C for 1h, 256 μl of 10% (wt/vol) sodium dodecyl sulfate (SDS) and 0.5 ml of TES containing 0.5 mg of protease K were added, and the mixture was incubated for 1 h at 65°C. Then, 1.8 ml of 10% SDS was added, and the mixture was incubated for another 1 h at 65°C. Total DNA was precipitated by adding 2.5 volumes of ethanol at room temperature. The DNA obtained by centrifugation at 6000 g for 10 min at room temperature was washed with 70% ethanol and dissolved in 1 ml TES for use.

The 16S rRNA gene-containing fragment of approximately 1.5 kb was amplified with the 27F (AGAGTTTGATCMTGGCTCAG) and 1492R (TACGGYTACCTTGTTACGACTT) primers using 2 μl DNA as the template, and the PCR products were sequenced [20]. PCR was done with a total volume of 50 μl, which contained 27.8 μl Sterilized $H_2O$, 5 μl 10×PCR Buffer, 5 μl dNTP Mix, 4μl $MgCl_2$, 3 μl Primer1, 3 μl Primer2, 0.2 μl Taq DNA Polymerase and 2 μl Template DNA. Amplification of the DNA was done for 35 cycles under the following conditions: denaturation, 95°C for 30 s; primer annealing, 56°C for 30 s; and primer ex tension, 72°C for 90 s. Miseq platform was used and the concentration of each sample was about 300ng μl$^{-1}$. The almost-complete 16S rDNA gene sequences of these isolates were obtained and used for initial BLAST searches in GenBank and for phylogenetic analyses. The sequences were aligned to their nearest phylogenetic neighbors using Genetyx software (GENETYX Corporation, Tokyo, Japan), and calculation of pairwise 16S rDNA gene sequence similarities was achieved using the EMBL sever (http://www.ebi.ac.uk/Tools/psa/). Phylogenetic trees were generated using same software. The robustness of the inferred topologies was tested by using 1,000 bootstrap re-samplings.

## Optimum growth conditions for temperature and NaCl concentration

Each bacterium was grown to the stationary phase in Marine Broth 2216 at 25°C. The resulting cells were collected by centrifugation (5000×g, 5 min) and resuspended in 100 ml of the ONR7a medium to an optical density at 600 nm ($OD_{600}$) of 0.1. The cultures were incubated at different temperatures (20, 30, 35, 40, and 45°C) and NaCl concentration (0 g $L^{-1}$, 15 g $L^{-1}$, 22.8 g $L^{-1}$, 40 g $L^{-1}$, and 50 g $L^{-1}$). A portion of the culture was collected and measured at $OD_{600}$ after being incubated with shaking. Each experiment was done in parallel three times.

## Degradation of different lignin-derived compounds by isolates

Each bacterium was grown to the stationary phase in Marine Broth 2216 at 25°C. The resulting cells were washed twice with ONR7a medium and then suspended in 5 ml of the same medium to give a turbidity of 0.05–0.1 at 600 nm. Each of the lignin-derived compounds (4-hydroxybenzoic acid, benzoic acid, ferulic acid and syringic acid) was added to a final concentration of 0.1%, and the mixture was shaken (150 rpm) at 30°C. Each experiment was done in one time.

## Degradation products of ferulic acid by isolates

Isolates were inoculated into ONR7a medium supplemented with ferulic acid and incubated for 24 h under conditions optimized. The concentrations of degradation products were determined with HPLC.

## Analysis of metabolism of lignin-derived compounds by HPLC and GC-MS

0.5 ml of the culture was collected and mixed with equal amounts of methanol. The mixture was filtered through a Millipore Filter Membrane (0.22 μm/50 mm) and then subjected to high-performance liquid chromatography (HPLC) analysis. An Alliance HPLC instrument (Waters, Milford, Mass.) with an octadecyl silica reverse-phase column (30 cm in length and 4.6 mm in diameter; Waters) was used at 40°C. A mobile phase consisting of solvent A (0.1% trifluoroacetic acid in water) and solvent B (acetonitrile) was used at a flow rate of 1 ml/min. After injection of 10 μl of a sample, 90% solvent A was run through the column for 4 min, after which the percentage of solvent B was gradually increased from 10% to 100% over 15 minutes. Finally, 100% of solvent B was run for 2 min. The absorbance spectrum of the eluate was monitored from 200 to 500 nm with a photodiode array detector (Waters 2998). The retention times of vanillic acid and ferulic acid were 6.237 and 7.332 min, respectively. All vanillic acid and ferulic acid we have used are standard. The test samples were extracted by ethyl acetate and dried over anhydrous $Na_2SO_4$ and the solvent was allowed to evaporate at 40°C. The residual was dissolved in an equal volume of methanol to original liquid sample and analyzed by gas chromatography-mass spectrometry (GC-MS) (Shimadzu QP5050A instrument) with a DB-5 column (length, 30 m; diameter, 0.25 mm) (J&W Scientific). The column temperature was initially kept at 50°C for 5 min before being increased to 300°C at a rate of 2.5°C per min. The injector and detector temperatures were both 300°C. Determination of vanillic acid, vanillin, vanillate, protocatechuate, and fumarate through SIM (selective reaction monitoring) with referenced ion fragments information as showed as following: 168.15m/z, 152.15m/z, 151.03m/z, 154.12m/z and 115.05m/z, respectively.

## Enzyme assays for PCA dioxygenase

Isolates were grown in 600 ml of Marine Broth for 12 h at 25°C with shaking and then washed twice with ONR7a. We centrifuged the pre-cultured bacterial solution (5000× g, 5 min),

removed the supernatant, then washed twice with seawater medium, removed the supernatant, and blotted dry with a pipette tip. Finally, the bacterial solution was dissolved in a seawater medium of 5 ml. The resulting cells were resuspended in 200 ml of ONR7a containing 0.2% vanillic acid, and the mixture was incubated with shaking at 25°C. The cells were harvested when approximately half of the vanillic acid was degraded, washed twice with 50 mM Tris-HCl (pH 8.0), and disrupted by ultrasonication. The amount of vanillic acid was determined by spectrophotometer. The cell-free extracts were obtained by centrifugation. The reaction mixture for assaying PCA dioxygenase activity consisted of 1900 μl of 50 mM Tris-HCl (pH 8.0), 40 μl of 1 mM PCA, and 60 μl of cell-free extract (100 μg ml$^{-1}$). The reaction mixture was incubated at 25°C and monitored spectrometrically from 200 to 500 nm with an EVOLUTION 260 BIO UV-Visible spectrophotometer (ThermoFisher, USA).

## Results

### Isolation of vanillic acid degraders

In this study, vanillic acid was used as lignin-derived low molecular weight compound to isolate vanillic acid-utilizing bacteria from marine and sediment samples collected from the coast of Lianyungang and the China East Sea, respectively. The predominant strains, which have adapted to their marine environment, could be isolated using the direct plating method, which resulted in isolation of bacteria with the highest growth rate in laboratory conditions. Three strains with the ability to use vanillic acid as the sole carbon and energy source were successfully isolated by direct plating, namely, strains ZXY12VA01, ZXY12VA14, and ZXY12VA15 (Table 1). Three additional strains with the ability to use vanillic acid as the sole carbon and energy source were isolated using the enrichment technique; namely, ZXY12VA05, ZXY12VA16, and ZXY12VA17.

The colony morphology of the isolates when plated with vanillic acid are as follows: ZXY12VA01, white and small; ZXY12VA05, milk yellow, uplifted with a smooth edge, large; ZXY12VA14, milk yellow, smooth edge, uplifted; ZXY12VA15, white and small, uplifted, rapid growth; ZXY12VA16, small, dark brown, slow growth, secreted brown pigment; ZXY12VA17, primrose yellow and turned to yellow. All strains are Gram-negative, except strain ZXY12VA05, which is Gram-positive (Fig 1).

Electrophoresis results of PCR products (1500 bp) of isolated bacteria are shown in S1 Fig. Phylogenetic analysis based on the neighbor-joining algorithm revealed that strain ZXY12VA01 clusters with the *Halomonas* genus (Fig 2). The most closely related type strain is *Halomonas titanicae* BH1$^T$ (99.87% 16S rDNA sequence similarity). Other related type strains are *Halomonas variabilis* DSM 3051$^T$ (99.0%), *Halomonas olivaria* C17$^T$ (99.0%), and *Halomonas neptunia* Eplume1$^T$ (99.0%). Based on these data, strain ZXY12VA01 likely belongs to *Halomonas*, which belongs to the family *Halomonadaceae*. This heterogeneous family, originally proposed by Frenzmann et al. (1988), to date includes more than 60 species [21]. They

**Table 1. Strains isolated in this study.**

| Strain name | Isolation method | Sampling site | Relative type strain | Identity (%) |
|---|---|---|---|---|
| ZXY12VA01 | Enrichment | Lianyungang | *Halomonas titanicae* BH1T(FN433898) | 99.87 |
| ZXY12VA05 | Direct plating | China East Sea | *Arthrobacter nicotianae* | 99.80 |
| ZXY12VA14 | Enrichment | China East Sea | *Pseudoalteromonas prydzensis* | 95.36 |
| ZXY12VA15 | Enrichment | China East Sea | *Marinomonas brasilensis* | 97.22 |
| ZXY12VA16 | Direct plating | Lianyungang | *Thalassospira profundimaris* | 99.85 |
| ZXY12VA17 | Direct plating | China East Sea | *Pseudoalteromonas nigrifaciens* | 99.98 |

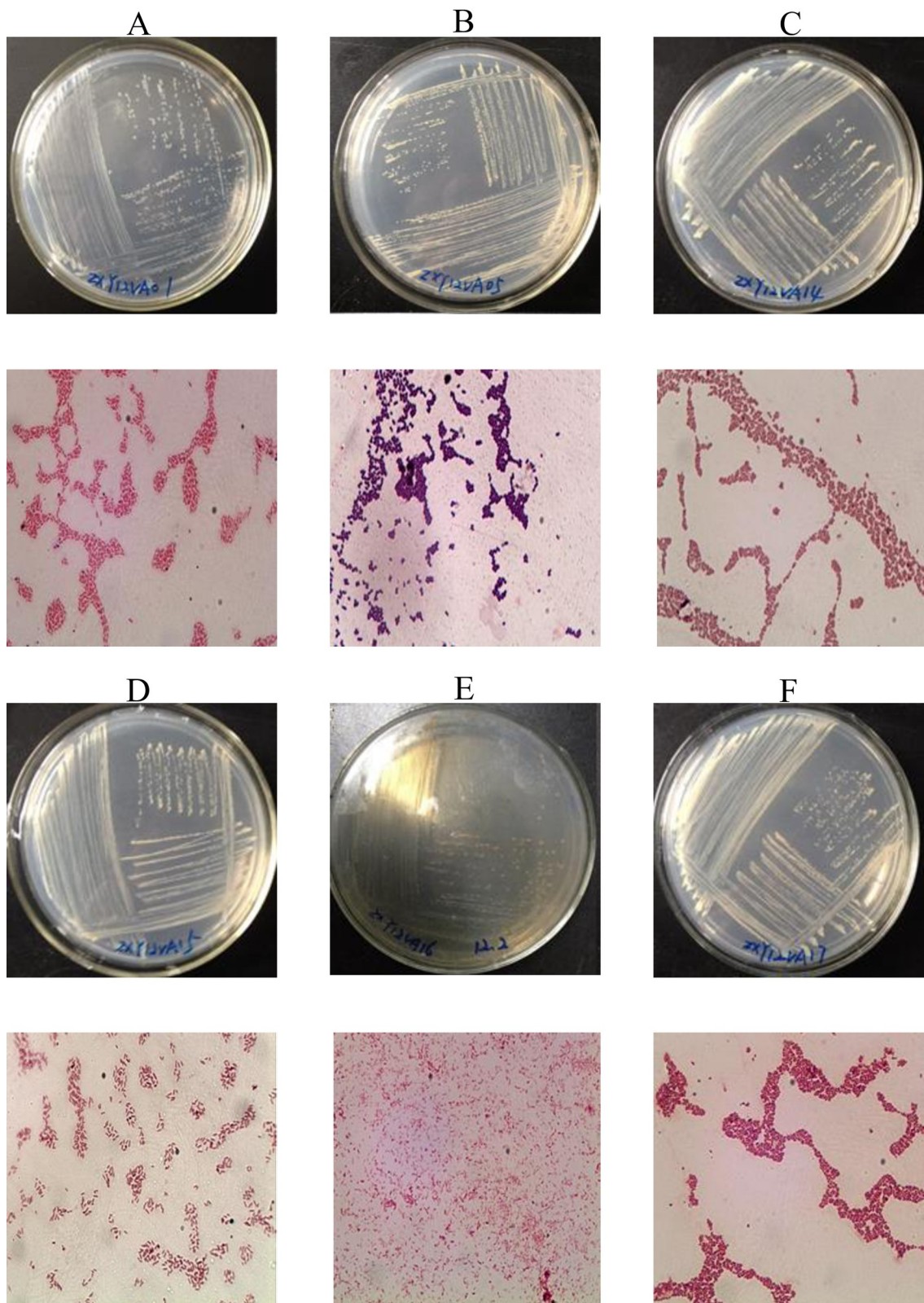

**Fig 1. Gram stain of isolates ZXY12VA01 (A), ZXY12VA05 (B), ZXY12VA14 (C), ZXY12VA15 (D), ZXY12VA16 (E), ZXY12VA17 (F).**

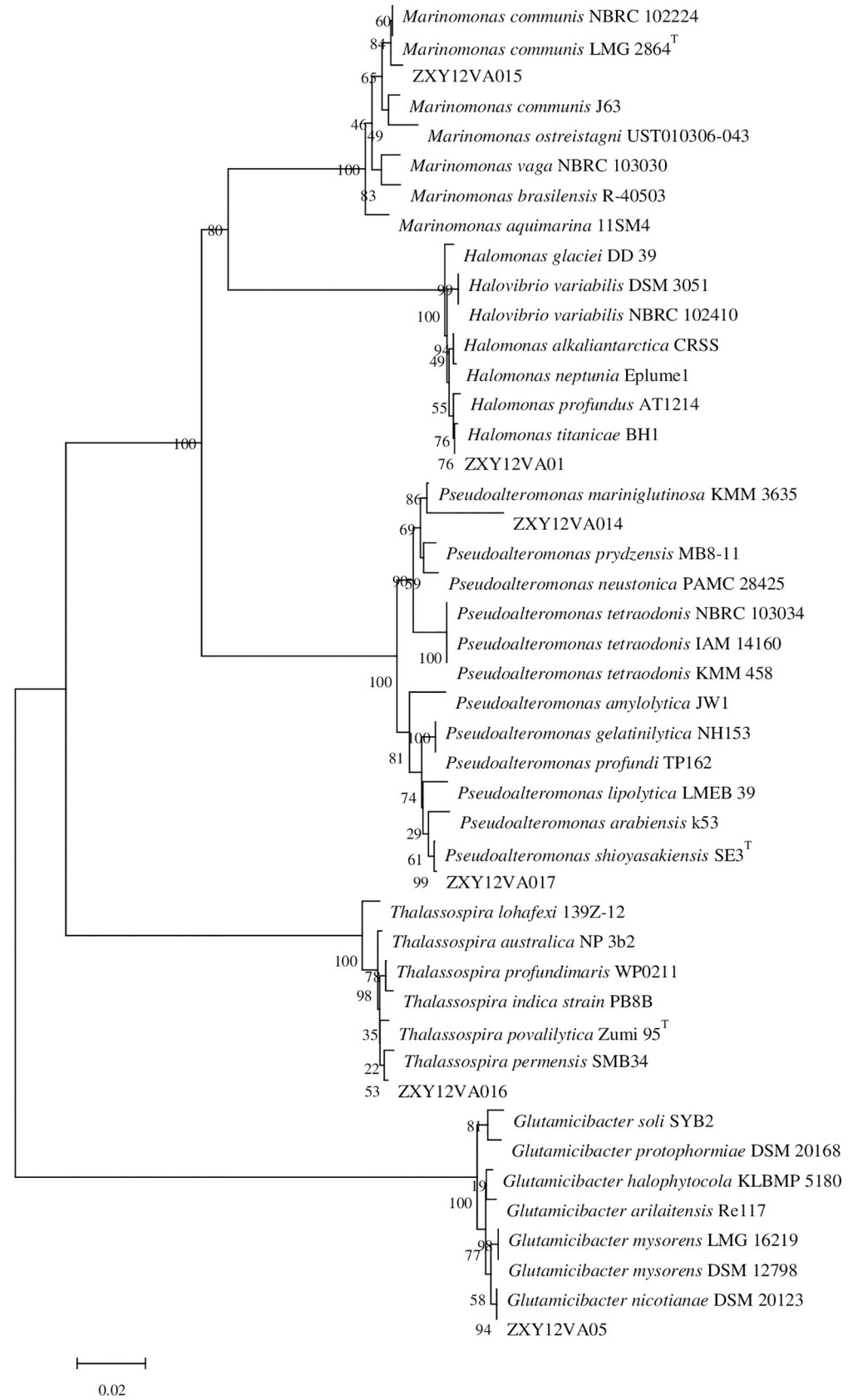

**Fig 2. Phylogenetic tree of thermophilic isolates based on 16S rDNA sequences.**

have been isolated from many water and soil environments within saline, hypersaline or alkaline habitats. *Halomonas titanicae* BH1[T] was isolated from rusticle samples collected at the site of the wreck using Bacto marine agar 2216 medium (Difco). It grows optimally at 30–37˚C in the presence of 2–8% (w/v) NaCl.

The most closely related type strain to ZXY12VA05 is *Arthrobacter nicotinanae* DSM 20123T (99.86%). Other related type strains include *Arthrobacter mysorens* LMG 16219[T], *Arthrobacter arilaitensis* Re117[T], and Arthrobacter uratoxydans DSM 20647[T]. Based on these data, strain ZXY12VA05 belongs to the genus *Arthrobacter*. Members of the *Arthrobacter* genus have a high G+C content, are actinomycete-coryneform bacteria, and are all Gram-positive. This genus contains more than 40 species of bacteria isolated from diverse environments, including soil and clinical specimens. Most *Arthrobacter* strains are mesophiles; however, some members have been isolated from Antarctic lakes and deep-sea sediment [22].

Both ZXY12VA14 and ZXY12VA17 strains cluster with the genus *Pseudoalteromonas* (Fig 2). The most closely related type strain for ZXY12VA14 and ZXY12VA17 are *Pseudoalteromonas mariniglutinosa* KMM 3635[T] (95%) and *Pseudoalteromonas shioyasakiensis* SE3[T] (98%), respectively. Other related type strains are *Pseudoalteromonas prydzensis* MB8-11[T] for ZXY12VA14; and *Pseudoalteromonas arabiensis* k53[T], *Pseudoalteromonas lipolytica* LMEB 39[T], and *Pseudoalteromonas byunsanensis* FR1199[T] for ZXY12VA17. Members of *Pseudoalteromonas* are characteristically Gram-negative, rod-shaped, motile bacteria with one or two polar flagella. Most *Pseudoalteromonas* strains are mesophiles; however, some halotolerant psychrophilic members have been isolated from Antarctic sea ice [23]. Strain KMM 3635[T] was isolated from a Chaetoceros lauderi diatom, which was collected from seawater in the Marseille Gulf [24]. Strain SE3T was isolated from Pacific Ocean sediment [25].

Phylogenetic analysis revealed that the most closely related type strain to ZXY12VA15 (Fig 2) is *Marinomonas communis* LMG 2864[T] (99.15% 16S rDNA sequence similarity). Other related type strains include *Marinomonas vaga* ATCC 27119[T], *Marinomonas aquimarina* CECT 5080[T], and *Marinomonas fungiae* AN44[T]. Based on these data, strain ZXY12VA15 belongs to the genus *Marinomonas*, which was created in 1983 and comprises 15 species. The members are Gram-negative, aerobic bacterium and have mainly been isolated from geographically diverse marine environments [26].

Strain ZXY12VA16 clusters with the *Thalassospira* genus (Fig 2). The most closely related type strain is *Thalassospira povalilytica* Zumi 95[T] (99.62% 16S rDNA sequence similarity). Other related type strains include *Thalassospira profundimaris* WP0211[T], *Thalassospira tepidiphila* 1-1BT, and Thalassospira permensis NBRC 106175[T]. Members of the *Thalassospira* genus are vibrioid to spiral cells; they are strict aerobes that are unable to grow under anaerobic conditions by fermentation, respiration or photoheterotrophy [27]. *Thalassospira povalilytica* Zumi 95[T] was isolated from plastic rope litter found in the Tokyo Bay, Japan, and exhibits the ability to utilize polyvinyl-alcohol as a carbon source [28].

The sequence data reported in this study appear in the EMBL, GenBank and DDBI sequence databases under the accession numbers. The GenBank accession number of strains ZXY12VA01, ZXY12VA05, ZXY12VA14, ZXY12VA15, ZXY12VA16 and ZXY12VA17 are KT274805, KT274806, KT717632, KT274807, KT274808 and KT274809, respectively.

## Optimum growth conditions for temperature and NaCl concentration

Marine and terrestrial environments are highly variable; however, microorganisms have the ability to rapidly evolve and adapt to changing environments. The optimum growth conditions of terrestrial bacteria in terms of temperature and NaCl concentration are generally 30˚C and 0.5%, respectively. In contrast, marine microorganisms prefer to grow at temperatures under

30°C and at a high NaCl concentration. Thus, by investigating the optimal temperature and salt concentrations of the isolates, it may be possible to infer if these organisms originated from the terrestrial or the marine environment.

The optimum growth temperature of the isolates rang from 30 to 40°C (Fig 3). The optimum growth temperature for strains ZXY12VA01, ZXY12VA15, ZXY12VA16, and ZXY12VA17 is 35°C; whereas strains ZXY12VA05 and ZXY12VA14 grow optimally at 30°C. These isolates do not grow above 45°C or below 20°C.

All isolates grow well in a wide NaCl concentration range from 0 to 50 g $L^{-1}$, with the optimum concentration being 22.8 g $L^{-1}$, which is the same as natural seawater (Fig 4). These 6 strains appear to have adapted to the NaCl concentration of seawater; therefore, they are not terrestrial microorganisms that were recently deposited into the marine environment.

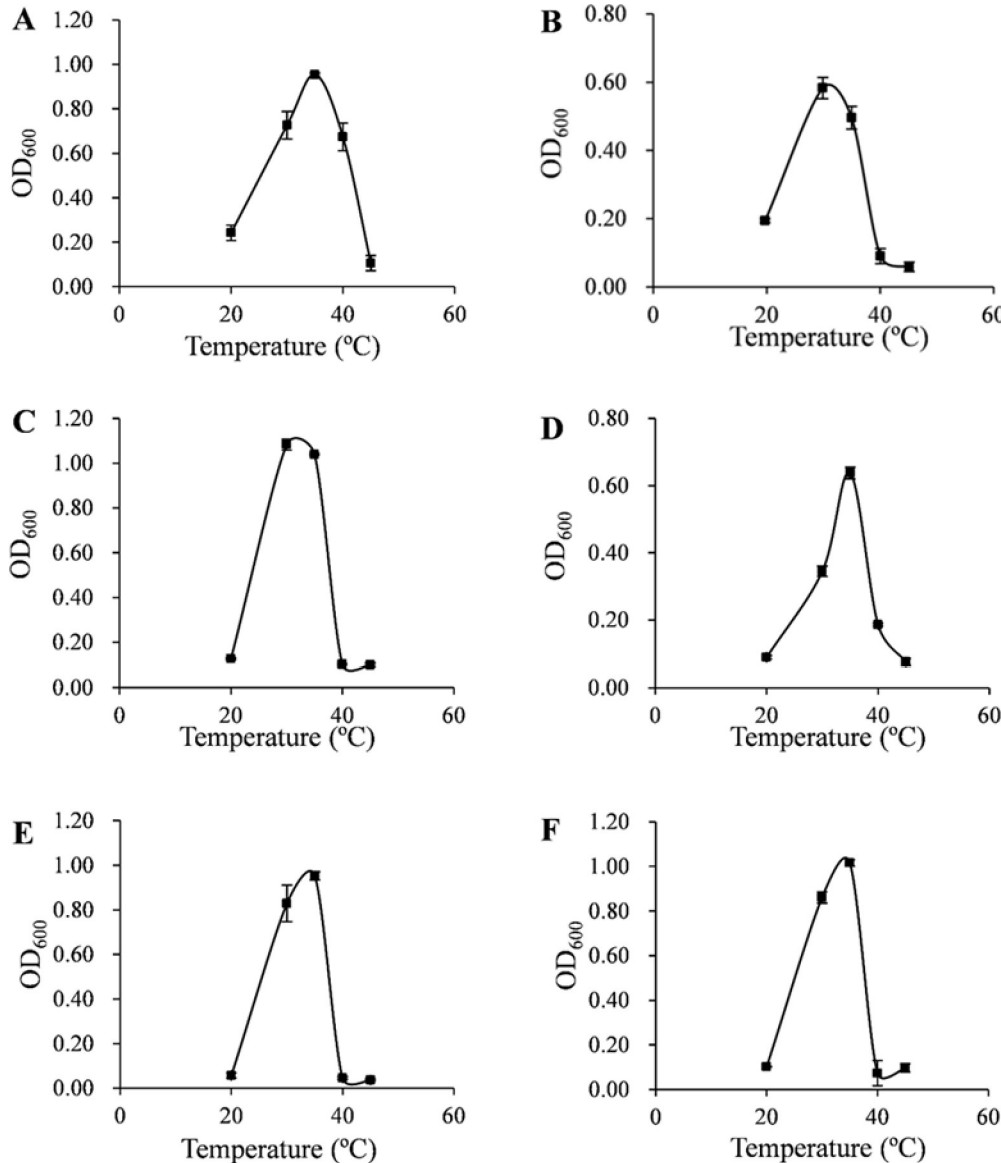

**Fig 3. Optimum growth temperatures of isolates ZXY12VA01 (A), ZXY12VA05 (B), ZXY12VA14 (C), ZXY12VA15 (D), ZXY12VA16 (E), ZXY12VA17 (F).**

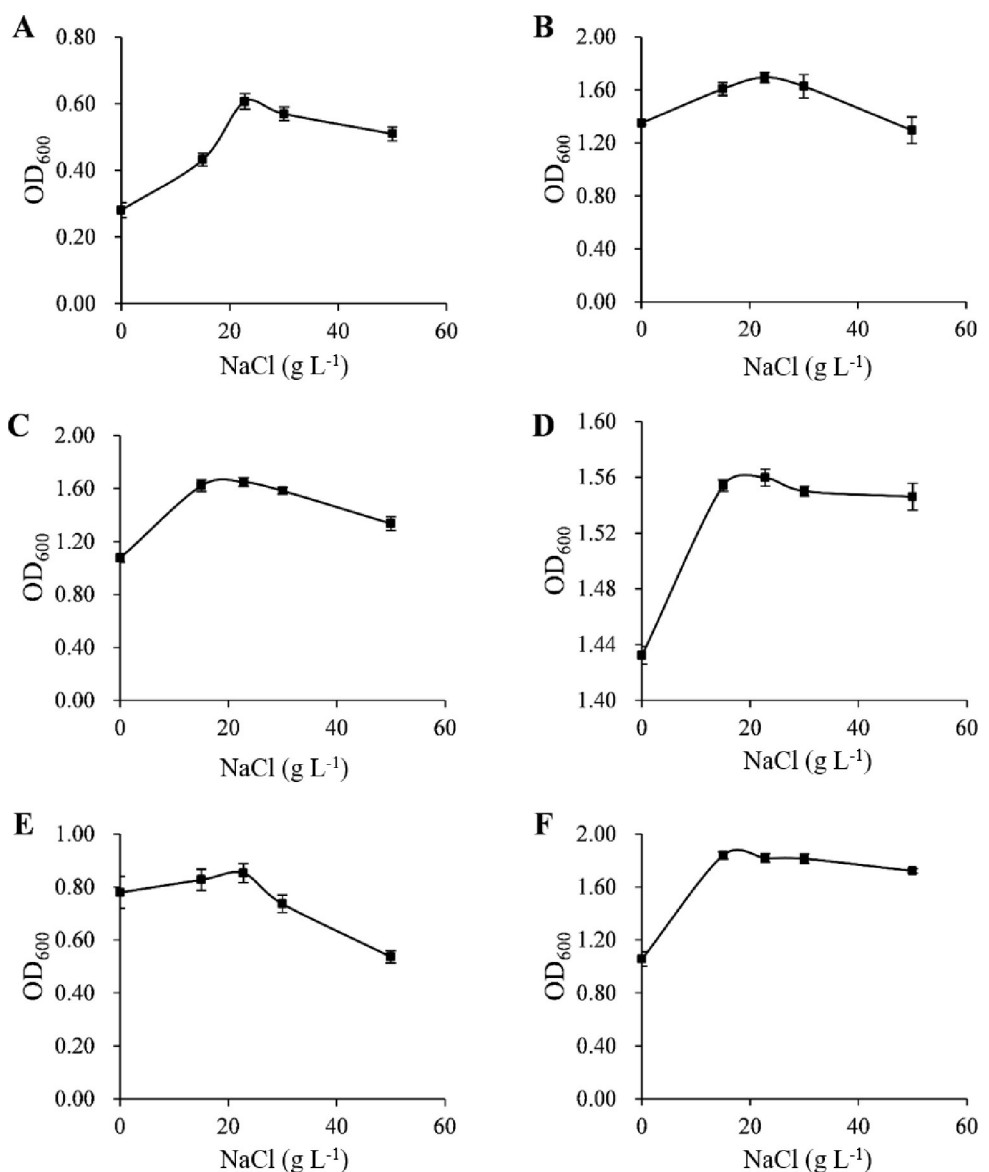

**Fig 4. Optimum growth in different NaCl concentrations of isolates ZXY12VA01 (A), ZXY12VA05 (B), ZXY12VA14 (C), ZXY12VA15 (D), ZXY12VA16 (E), and ZXY12VA17 (F).**

## Ability to utilize lignin-derived compounds

In addition to vanillic acid, it is possible for some organisms to utilize other lignin-derived low molecular weight compounds, such as benzoic acid, 4-hydroxybenzoic acid, ferulic acid, and syringic acid. Thus, the isolates were cultured for 10 days at 30°C in the selected lignin-derived compounds to test their ability to use these compounds as their sole carbon and energy source. 4-Hydroxybenzoic acid was the best substrate for all isolates because it resulted in cell densities of approximately 1.0 of $OD_{600}$ (Table 2). The cell densities grown on syringic acid and ferulic acid were much lower than those on 4-hydroxybenzoic acid; however, strains ZXY12VA05, ZXY12VA14, ZXY12VA16 grew as well on benzoic acid as on 4-hydroxybenzoic acid, while ZXY12VA01 and ZXY12VA15 did not. These results demonstrate that the strains isolated in

**Table 2. Vanillate-degrading isolates grown on different lignin-derived carbon sources.**

| Strain name | Carbon source | $OD_{600}$ | | | | |
|---|---|---|---|---|---|---|
| | | 1 d | 2 d | 3 d | 6 d | 10 d |
| ZXY12VA01 | Benzoate | 0.068 | 0.551 | – | – | – |
| | 4-Hydroxybenzoate | 1.061 | – | – | – | – |
| | Ferulate | 0.281 | 0.314 | – | – | – |
| | Syringate | 0.241 | 0.283 | – | – | – |
| ZXY12VA05 | Benzoate | 0.138 | – | 1.222 | – | – |
| | 4-Hydroxybenzoate | 1.325 | – | – | – | – |
| | Ferulate | 0.355 | – | 0.392 | – | – |
| | Syringate | 0.298 | – | 0.307 | – | – |
| ZXY12VA14 | Benzoate | – | – | – | – | 1.232 |
| | 4-Hydroxybenzoate | 1.992 | – | – | – | – |
| | Ferulate | – | – | – | – | 0.45 |
| | Syringate | – | – | – | – | 0.316 |
| ZXY12VA15 | Benzoate | – | – | – | 0.741 | – |
| | 4-Hydroxybenzoate | 0.941 | – | – | – | – |
| | Ferulate | – | – | – | 0.392 | 0.443 |
| | Syringate | – | – | – | 0.307 | 0.334 |
| ZXY12VA16 | Benzoate | – | – | – | 1.172 | – |
| | 4-Hydroxybenzoate | 1.022 | – | – | – | – |
| | Ferulate | 0.336 | – | – | – | 0.524 |
| | Syringate | 0.366 | – | – | – | 0.523 |
| ZXY12VA17 | Benzoate | – | – | 1.102 | – | – |
| | 4-Hydroxybenzoate | – | – | 0.952 | – | – |
| | Ferulate | – | – | 0.420 | – | 0.447 |
| | Syringate | – | – | 0.290 | – | 0.439 |

this study have the ability to grow on several lignin-derived compounds and likely play an important role in lignin mineralization in marine environments.

## Detection of ferulic acid intermediates

Ferulic acid, an easily available component of lignin, plays an important role in the cross-linking of cell walls in various plants. The first step in ferulic acid degradation is the removal of its side chain. Two types of side chain removal have been reported for a number of microorganisms [29]. One type, which results in a 4-hydroxy-3-methoxystyrene intermediate, is catalyzed by a non-oxidative decarboxylase that eliminates one carbon from the ferulic acid side chain. The 4-hydroxy-3-methoxystyrene is subsequently transformed to vanillin and then to vanillic acid. The other type, which results in the direct formation of vanillin, involves the elimination of two carbons from the ferulic acid side chain. Vanillin is subsequently oxidized to vanillic acid by vanillin dehydrogenase.

To determine which type of metabolism is carried out by these strains, metabolites from cultures grown with ferulic acid as the sole carbon source were investigated using HPLC. Vanillic acid was detected in all culture media when isolates were grown on ferulic acid as the sole carbon source; however, no 4-hydroxy-3-methoxystyrene was detected (Figs 5 and 6), indicating that ferulic acid metabolism by these strains occurs via the elimination of two side chain carbons. Through this study, we found that ZXY12VA16 produced the most amount of

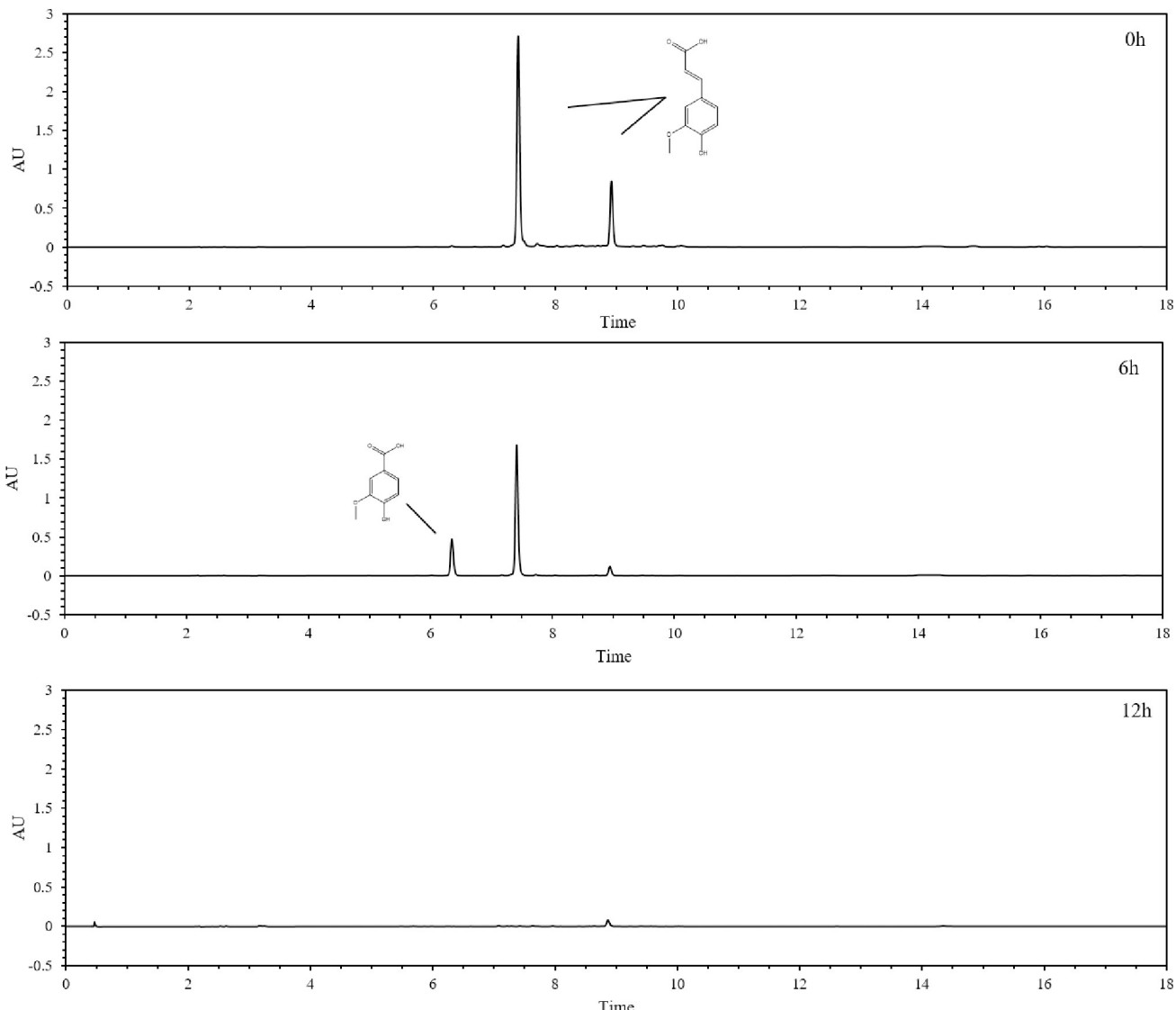

**Fig 5. The formation of the intermediates in the biodegradation of ferulic acid in ZXY12VA01 shown in the HPLC analysis.** The major metabolites were detected on the 0h (A), 12h (B) and 24h (C), respectively.

vanillic acid. The ZXY12VA15 strain is a dominant strain of aromatic degrading bacteria. Normally, feruloyl-CoA synthetase and enoyl-CoA hydratase/aldolase are found in most FA-degrading strains. These strains convert ferulic acid to vanillin by a non-β-oxidative pathway [30]. It demonstrates that vanillic acid is an important intermediate in the metabolism of lignin and plays a leading role in the metabolism of aromatic compounds.

When strain ZXY12VA01, ZXY12VA05 and ZXY12VA14 was grown on 0.1% vanillic acid as the sole carbon, the formation of vanillin and protocatechuic acid was detected in the culture medium (Fig 7). Vanillic acid can be hydroformylated to form vanillin or demethylated to form protocatechuic acid. Protocatechuic acid is considered to be the target compound for ring fission. Protocatechuate has a vic-diol structure, which can be subjected to ring cleavage by dioxygenase. The formation of fumaric acid proves that the metabolite is finally decomposed into the tricarboxylic acid cycle (Fig 8).

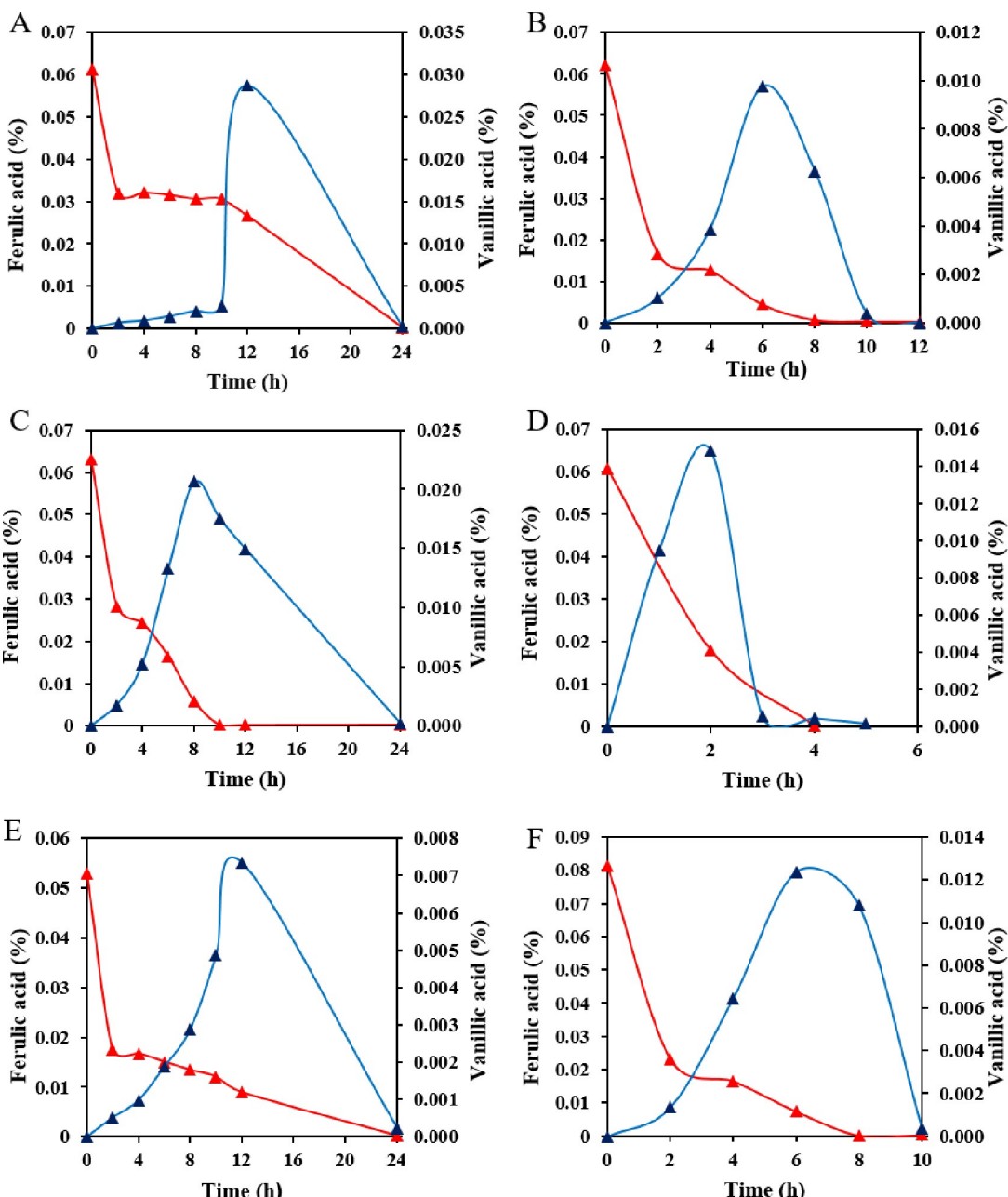

**Fig 6. Metabolism of ferulic acid to vanillic acid by isolates ZXY12VA01 (A), ZXY12VA05 (B), ZXY12VA14 (C), ZXY12VA15 (D), ZXY12VA16 (E), ZXY12VA17 (F).**

### Enzyme assay for aromatic ring-fission reaction

Vanillic acid metabolism involves removal of the methoxy group by a demethylase, resulting in the formation of a protocatechuic acid (PCA) intermediate. Ring cleavage of PCA occurs by three known dioxygenases: PCA 2,3-dioxygenase, PCA 3,4-dioxygenase, and PCA 4,5-dioxygenase and results in the ring-cleavage products 5-carboxy-2-hydroxymuconate-6-semialdehyde (350 nm), 5-carboxymuconate (290 nm), and 4-carboxy-2-hydroxymuconate-6-semialdehyde (410 nm), respectively. Each ring-fission product has a specific absorption peak, making it easy to determine which ring-cleavage occurred.

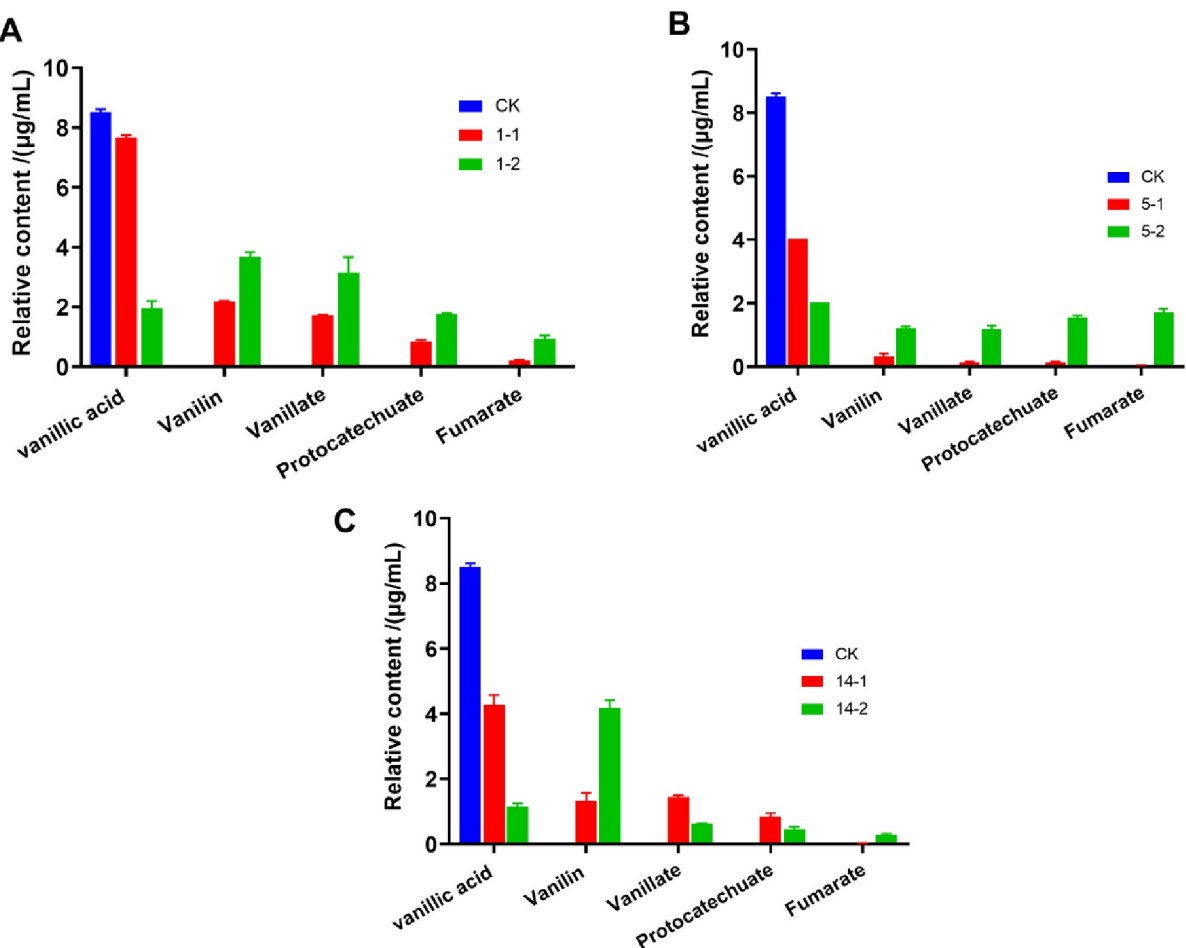

**Fig 7. Growth of ZXY12VA01 (A), ZXY12VA05 (B) and ZXY12VA14 (C) on vanillic acid. The four major metabolites were detected on the 12h and 24h, respectively.**

To induce vanillic acid-metabolizing enzymes, ZXY12VA15 was grown on vanillic acid and harvested when approximately half of the carbon source was degraded. The cell-free extracts were applied to the enzyme assay using PCA as the substrate. Spectrophotometric assays provided evidence for the formation of 4-carboxy-2-hydroxymuconate-6-semialdehyde ($\lambda_{max}$, 410 nm) at pH 8.0 (Fig 9). It was determined that strain ZXY12VA15 were capable of 4,5-cleavage.

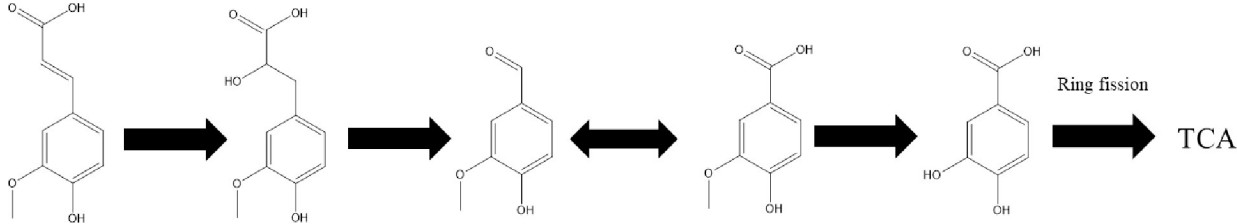

**Fig 8. Speculated biodegradation pathways of ferulic acid and vanillic acid by marine bacteria.**

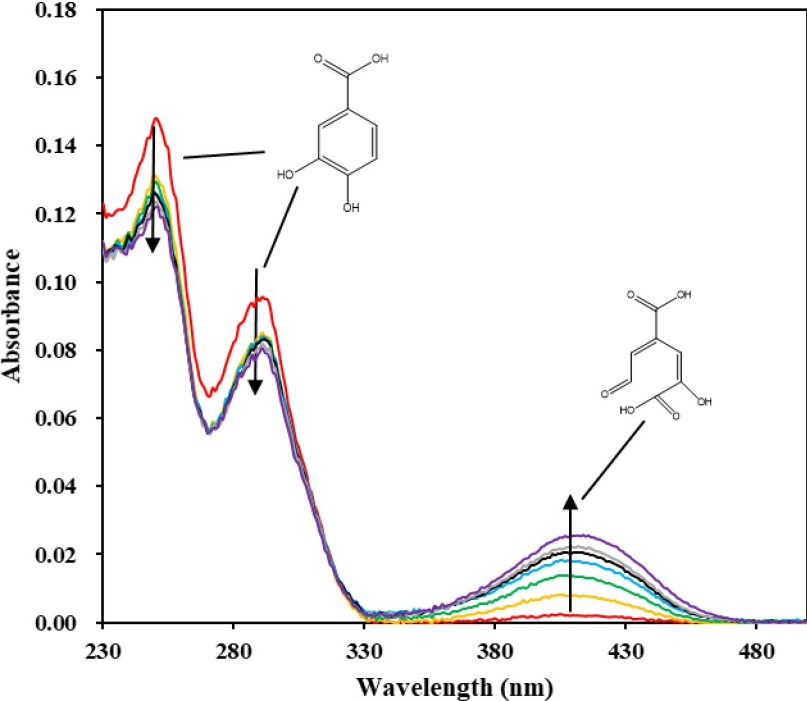

**Fig 9. Spectrophotometric analysis of ZXY12VA15 enzyme activity.** Absorbance increase of 4-carboxy-2-hydroxymuconate-6-semialdehyde ($\lambda_{max}$, 410 nm).

## Discussion

Genome-wide surveys were performed on the isolate representative species of related genera to identify genes associated with the PCA ring-cleavage pathway and the genomic information of strains were searched on KEGG.

*Halomonas elongate* DSM 2581 is a halotolerant bacteria isolated from a solar salt facility. It synthesizes the compatible solute Ectoine, thus allowing it to withstand NaCl concentrations well above 10% [31]. Contained in the *H. elongata* genome is a gene cluster (HELO_3955–3963) encoding PCA 3,4-dioxygenase and additional enzymes involved in the metabolism of ring-cleavage products.

*Marinomonas mediterranea* MMB-1 [32], *Marinomonas* sp. MWYL1 [33, 34], and *Marinomonas posidonica* IVIA-Po-181 [35] were isolated from marine samples. Their genomes have the same gene cluster, which encodes for PCA 4,5-dioxygenase and enzymes involved in the ring-cleavage product metabolism.

The members of *Arthrobacter* are among the most frequently isolated, indigenous, aerobic genera found in soils. *Arthrobacter phenanthrenivorans* Sph3 was isolated from a creosote-polluted site, where a wood-preserving factory had operated for over 30 years. It is able to metabolize phenanthrene as the sole carbon and energy source [36]. *Arthrobacter aurescens* TC1 was originally isolated from soil at an atrazine spill site [37]. *Arthrobacter chlorophenolicus* A6, a Gram-positive Actinobacterium, was enriched from a soil slurry using increasing concentrations of 4-chlorophenol and can degrade exceptionally high concentrations of 4-chlorophenol (up to 2.7 mM), along with other *p*-substituted phenols, including 4-nitrophenol and 4-bromophenol [38]. *Arthrobacter arilaitensis* Re117 was isolated from the surface of cheeses and possesses a salt-tolerance system [39]. *Arthrobacter* sp. FB24 was isolated from a soil sample that was obtained at a site contained by mixed waste that included both petroleum

hydrocarbons and extremely high metal levels. The genomes of these *Arthrobacter* members have the same gene cluster encoding PCA 3,4-dioxygenase and enzymes involved in the ring-cleavage product metabolism.

*Thalassospira xiamenensis* M-5 (DSM17429) was isolated from an oil-waste pool at an oil storage dock. Strain M-5 has a gene cluster encoding PCA 3,4-dioxygenase and enzymes involved in the ring-cleavage product metabolism [40].

*Pseudoalteromonas atlantica* T6c is a biofilm-forming, Gram-negative, motile marine bacterium. It is chemoorganotrophic and secrets extracellular enzymes that are able to hydrolyze agar, alginate, and carrageenan [41]. Strain T6c has a gene cluster encoding PCA 4,5-dioxygenase and enzymes involved in the ring-cleavage product metabolism.

This study demonstrates that members of the genera *Halomonas*, *Arthrobacter*, *Pseudoalteromonas*, *Marinomonas*, and *Thalassospira* are involved in the degradation of lignin-derived compounds in marine environments. In addition, many environmental pollutants are aromatic compounds, and marine bacteria capable of degrading lignin can also degrade most environmental pollutants. The isolated lignin-degrading bacteria can be applied to environmental pollution treatment in a marine environment or a high-salt environment. Although we have isolated marine bacteria that can degrade vanillic acid, our further research will focus on applying to actual production.

## Conclusion

In this study, 6 bacterial strains were isolated from marine samples using the lignin-derived compound vanillic acid as the sole carbon and energy source. It is concluded that these 6 strains have adapted to the NaCl concentration of seawater and are marine dominant bacteria widely present in the marine environment, rather than the soil microorganisms of the recent land. They are Phylogenetic analysis indicates that ZXY12VA01, ZXY12VA05, ZXY12VA14, ZXY12VA15, ZXY12VA16, and ZXY12VA17 are members of *Halomonas*, *Arthrobacter*, *Pseudoalteromonas*, *Marinomonas*, and *Thalassospira*, respectively. They are also able to use other lignin-derived compounds, such as 4-hydroxybenzoic acid, ferulic acid, syringic acid, and benzoic acid. The isolates showed 3,4-dioxygenase or 4,5-dioxygenase activity for protocatechuic acid ring-cleavage, which is consistent with information on related genera genome sequences.

## Supporting information

**S1 Fig. Electrophoresis image of 16S rDNA PCR products.**
(DOCX)

## Author Contributions

**Conceptualization:** Peng Lu, Weinan Wang, Mengjiao Cao, Bo Yuan, Zhaozhong Feng.

**Data curation:** Peng Lu, Weinan Wang, Xiaoyan Zhang, Bo Yuan, Zhaozhong Feng.

**Formal analysis:** Weinan Wang, Zhaozhong Feng.

**Funding acquisition:** Peng Lu, Weinan Wang, Mengjiao Cao, Bo Yuan, Zhaozhong Feng.

**Investigation:** Anjie Jiang, Mengjiao Cao.

**Methodology:** Anjie Jiang.

**Validation:** Guangxi Zhang, Wen Li, Xue Peng.

**Visualization:** Guangxi Zhang, Ke Xing, Xue Peng, Bo Yuan, Zhaozhong Feng.

**Writing – original draft:** Weinan Wang, Zhaozhong Feng.

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
