## [Decision Letter · Decision Letter 0]

11 Mar 2020

PONE-D-19-35662

Isolation and Characterization Marine Bacteria Capable of Degrading Lignin-derived Compounds

PLOS ONE

Dear Mr. Feng,

Thank you for submitting your manuscript to PLOS ONE. After careful consideration, we feel that it has merit but does not fully meet PLOS ONE’s publication criteria as it currently stands. Therefore, we invite you to submit a revised version of the manuscript that addresses the points raised during the review process.

We would appreciate receiving your revised manuscript by Apr 25 2020 11:59PM. To enhance the reproducibility of your results, we recommend that if applicable you deposit your laboratory protocols in protocols.io, where a protocol can be assigned its own identifier (DOI) such that it can be cited independently in the future. For instructions see: http://journals.plos.org/plosone/s/submission-guidelines#loc-laboratory-protocols

We look forward to receiving your revised manuscript.

Kind regards,

Pankaj Kumar Arora

Academic Editor

PLOS ONE

Journal Requirements:

2) In your Methods section, please provide additional information regarding the permits you obtained for the work. Please ensure you have included the full name of the authority that approved the field site access and, if no permits were required, a brief statement explaining why.

3) Please ensure that you refer to Figure 4 in your text as, if accepted, production will need this reference to link the reader to the figure.

4) Please include captions for your Supporting Information files at the end of your manuscript, and update any in-text citations to match accordingly. Please see our Supporting Information guidelines for more information: http://journals.plos.org/plosone/s/supporting-information.

Reviewers' comments:

Reviewer's Responses to Questions

**Comments to the Author**

1. Is the manuscript technically sound, and do the data support the conclusions?

Reviewer #1: Yes

2. Has the statistical analysis been performed appropriately and rigorously? 

Reviewer #1: Yes

3. Have the authors made all data underlying the findings in their manuscript fully available?

Reviewer #1: Yes

4. Is the manuscript presented in an intelligible fashion and written in standard English?

Reviewer #1: Yes

5. Review Comments to the Author

Reviewer #1: The paper is well written and reports the new knowledge on the lignin degrading marine bacteria isolated from East China Sea. Following comments should be taken into account before its consideration for publication elsewhere:

1. Remove the word "Isolation" from the list of keywords.

2. Revised sentence as "organic matter into the oceans" at Line No. 21.

3. It should be "sediments" instead of "ediment" at Line No. 93.

4. It should be revised as "East China Sea in Zhoushan, Zhejiang (30°58′20.71″N, 122° 95 41′22.10″E), People's Republic of China" at Line No. 94-95.

5. The sentence should be started with "The depth of sediments" instead of "Depth" at Line No. 95.

6. Revised this sentence as "Further, the sediments samples were placed in sealed plastic boxes surrounded with ice and brought to the laboratory within 24h" instead of "Placed in sealed plastic boxes surrounded with ice, the samples

97 were taken to the laboratory within 24h" at Line No. 95.

7. Kindly mention the a tentative purity of the Chinese chemicals within bracket at Line No. 101.

8. Kindly check the presence of reference both in the text as well as in the reference section.

9. Kindly also incorporate the HP-LC and GC-MS chromatogram in the manuscript.

6. PLOS authors have the option to publish the peer review history of their article (what does this mean?). If published, this will include your full peer review and any attached files.

Reviewer #1: No

---

## [Author Response · Author response to Decision Letter 0]

4 Jun 2020

Dear editors and reviewers:

Thank you for your letter and the reviewers’ comments on our manuscript entitled " Isolation and Characterization Marine Bacteria Capable of Degrading Lignin-derived Compounds" (PONE-D-19-35662). Those comments are very helpful for revising and improving our paper, as well as the important guiding significance to other research. We have studied the comments carefully and made corrections which we hope meet with approval. The main corrections are in the manuscript and the responds to the reviewers’ comments are as follows.

Replies to the editors’ comments:

Response：Thank you for your suggestions and we have reorganized the manuscript to meet PLOS ONE's style requirements.

Response：Thank you for your valuable comments. All experiments in the Methods were conducted according to protocols approved by Jiangsu Normal University and Anhui Normal University and completed by researchers of the manuscript. No additional authorization information is required for this work.

3. Please ensure that you refer to Figure 4 in your text as, if accepted, production will need this reference to link the reader to the figure.

Response：Fig 4 has been cited and all figures have been reordered and checked.

4. Please include captions for your Supporting Information files at the end of your manuscript, and update any in-text citations to match accordingly.

Response：Captions of Supporting Information has been added and rematched.

Replies to the reviewers’ comments:

1. Remove the word "Isolation" from the list of keywords.

Response：The word has been removed.

2. Revised sentence as "organic matter into the oceans" at Line No. 21.

Response：The sentence has been modified.

3. It should be "sediments" instead of "ediment" at Line No. 93.

Response：The word has been modified.

4. It should be revised as "East China Sea in Zhoushan, Zhejiang (30°58′20.71″N, 122° 95 41′22.10″E), People's Republic of China" at Line No. 94-95.

Response：The full name of China has been added, which is very important.

5. The sentence should be started with "The depth of sediments" instead of "Depth" at Line No. 95.

Response：We have modified it according to your suggestion.

6. Revised this sentence as "Further, the sediments samples were placed in sealed plastic boxes surrounded with ice and brought to the laboratory within 24h" instead of "Placed in sealed plastic boxes surrounded with ice, the samples were taken to the laboratory within 24h" at Line No. 95.

Response：We have modified it according to your suggestion.

7. Kindly mention the a tentative purity of the Chinese chemicals within bracket at Line No. 101.

Response：The purity of all chemicals has been added in the method. Thank you for your suggestion.

8. Kindly check the presence of reference both in the text as well as in the reference section.

Response：References have been checked and re-marked.

9. Kindly also incorporate the HP-LC and GC-MS chromatogram in the manuscript.

Response：Thanks for your kindly and profssional comment. In this manuscript, Fig 5 shows the analysis of FA degradation by HPLC. We used SIM (selective reaction monitoring) mode to determinated the content variation of vanillic acid, vanillin, vanillate, protocatechuate, and fumarate. In this method, five referenced ion fragments information were monitored and determined the content through calculation of ionic strength. We have reflected upon your comment and addiion of the description of method in the manuscript. 

Once again, thank you very much for your constructive comments and suggestions which would help us both in English and in depth to improve the quality of the paper.

Kind regards,

 Zhaozhong Feng

 E-mail: fzz2012@jsnu.edu.cn

---

## [Decision Letter · Decision Letter 1]

23 Jul 2020

PONE-D-19-35662R1

Isolation and Characterization Marine Bacteria Capable of Degrading Lignin-derived Compounds

PLOS ONE

Dear Dr. Feng,

Thank you for submitting your manuscript to PLOS ONE. After careful consideration, we feel that it has merit but does not fully meet PLOS ONE’s publication criteria as it currently stands. Therefore, we invite you to submit a revised version of the manuscript that addresses the points raised during the review process.

We look forward to receiving your revised manuscript.

Kind regards,

Pankaj Kumar Arora

Academic Editor

PLOS ONE

Reviewers' comments:

Reviewer's Responses to Questions

**Comments to the Author**

1. If the authors have adequately addressed your comments raised in a previous round of review and you feel that this manuscript is now acceptable for publication, you may indicate that here to bypass the “Comments to the Author” section, enter your conflict of interest statement in the “Confidential to Editor” section, and submit your "Accept" recommendation.

Reviewer #1: (No Response)

Reviewer #2: (No Response)

2. Is the manuscript technically sound, and do the data support the conclusions?

Reviewer #1: No

Reviewer #2: Partly

3. Has the statistical analysis been performed appropriately and rigorously? 

Reviewer #1: Yes

Reviewer #2: N/A

4. Have the authors made all data underlying the findings in their manuscript fully available?

Reviewer #1: Yes

Reviewer #2: Yes

5. Is the manuscript presented in an intelligible fashion and written in standard English?

Reviewer #1: Yes

Reviewer #2: Yes

6. Review Comments to the Author

Reviewer #1: (No Response)

Reviewer #2: There has been limited work done to characterize marine microorganisms that are able to aerobically degrade lignin-derived compounds. This study describes the process of isolating and characterizing several strains that can degrade these compounds. The isolates were evaluated for their substrate range and enzyme activity. The authors addressed each point in their response to reviewers letter. There are, however, some points in the manuscript that require further clarification, especially with respect to the methods and typographical errors. Please see my specific comments below.

Specific comments:

Line 88 – Contribute

Line 115 – The depth of the sediment….

136 – Vanillic acid as the sole carbon source

Lines 139-140- Please clarify your methods for enrichment. It is unclear what “this process” is referring to.

Line 156- Please refer to the specific primer (ie 27 Forward) in your PCR recipe.

Experiments described in Lines 179-188, 217 – What were your controls? How many replicates were used?

Line 207- Please include more details about your GC-MS preparation, such as extraction protocol and if the samples were derivatized.

Line 355 – Please clarify why the table shows OD values on a time scale of up to 10 days if the text is stating that the cultures were grown for 3 days with each substrate.

7. PLOS authors have the option to publish the peer review history of their article (what does this mean?). If published, this will include your full peer review and any attached files.

Reviewer #1: No

Reviewer #2: No

---

## [Author Response · Author response to Decision Letter 1]

16 Sep 2020

Dear editors and reviewers:

Thank you for your letter and the reviewers’ comments on our manuscript entitled " Isolation and Characterization Marine Bacteria Capable of Degrading Lignin-derived Compounds" (PONE-D-19-35662). Those comments are very helpful for revising and improving our paper, as well as the important guiding significance to other research. We have studied the comments carefully and made corrections which we hope meet with approval. The main corrections are in the manuscript and the responds to the reviewers’ comments are as follows.

Replies to the reviewers’ comments:

1. Line 88 – Contribute

Response：The word has been modified.

2. Line 115 – The depth of the sediment….

Response：The sentence has been modified.

3. 136 – Vanillic acid as the sole carbon source

Response：The sentence has been modified.

4. Lines 139-140- Please clarify your methods for enrichment. It is unclear what “this process” is referring to.

Response：“This process” is referring to the enrichment technique. The enrichment technique was performed in liquid ONR7a media amended with 0.2% vanillic acid. Cultures were incubated with shaking at 25 ºC until turbid.

5. Line 156- Please refer to the specific primer (ie 27 Forward) in your PCR recipe.

Response：The reference cited in the primer has been corrected.

6. Experiments described in Lines 179-188, 217 – What were your controls? How many replicates were used?

Response：Each experiment is an independent experiment, which reflects the content of metabolites according to the time course.

7. Line 207- Please include more details about your GC-MS preparation, such as extraction protocol and if the samples were derivatized.

Response：The test samples were extracted by ethyl acetate and dried over anhydrous Na2SO4 and the solvent was allowed to evaporate at 40℃. The residual was dissolved in an equal volume of methanol to original liquid sample and analyzed by GC-MS. Determination of vanillic acid, vanillin, vanillate, protocatechuate, and fumarate through SIM (selective reaction monitoring) with referenced ion fragments information. All samples were not derivatized.

8. Line 355 – Please clarify why the table shows OD values on a time scale of up to 10 days if the text is stating that the cultures were grown for 3 days with each substrate.

Response：We have modified it and the cultures were grown for 10 days with each substrate.

Once again, thank you very much for your constructive comments and suggestions which would help us both in English and in depth to improve the quality of the paper.

Kind regards,

Zhaozhong Feng

E-mail: fzz2012@jsnu.edu.cn

---

## [Editor Report · Decision Letter 2]

22 Sep 2020

Isolation and Characterization Marine Bacteria Capable of Degrading Lignin-derived Compounds

PONE-D-19-35662R2

Dear Dr. Feng,

We’re pleased to inform you that your manuscript has been judged scientifically suitable for publication and will be formally accepted for publication once it meets all outstanding technical requirements.

Kind regards,

Pankaj Kumar Arora

Academic Editor

PLOS ONE
---

## [Editor Report · Acceptance letter]

28 Sep 2020

PONE-D-19-35662R2 

Isolation and Characterization Marine Bacteria Capable of Degrading Lignin-derived Compounds 

Dear Dr. Feng:

I'm pleased to inform you that your manuscript has been deemed suitable for publication in PLOS ONE. Congratulations! Your manuscript is now with our production department. 

Kind regards, 

on behalf of

Dr. Pankaj Kumar Arora 

Academic Editor

PLOS ONE